complexity

COVID-19, transmission networks,
human mobility, migration, nosocomial infections

**Author for correspondence:**
Marian-Gabriel Hâncean
e-mail: gabriel.hancean@sas.unibuc.ro

# Early spread of COVID-19 in Romania: imported cases from Italy and human-to-human transmission networks

Marian-Gabriel Hâncean[1], Matjaž Perc[2,3,4]
and Jürgen Lerner[5]

[1]Department of Sociology, University of Bucharest, 90-92 Panduri Street, Bucharest 050663, Romania
[2]Faculty of Natural Sciences and Mathematics, University of Maribor, Koroška cesta 160, 2000 Maribor, Slovenia
[3]Department of Medical Research, China Medical University Hospital, China Medical University, Taichung 404, Taiwan
[4]Complexity Science Hub Vienna, Josefstädterstraße 39, 1080 Vienna, Austria
[5]Department of Computer and Information Science, University of Konstanz, Universitaetsstrasse 10, 78464 Konstanz, Germany

M-GH, 0000-0001-9358-5287; MP, 0000-0002-3087-541X;
JL, 0000-0001-6565-1873

We describe the early spread of the novel coronavirus (COVID-19) and the first human-to-human transmission networks, in Romania. We profiled the first 147 cases referring to sex, age, place of residence, probable country of infection, return day to Romania, COVID-19 confirmation date and the probable modes of COVID-19 transmissions. Also, we analysed human-to-human transmission networks and explored their structural features and time dynamics. In Romania, local cycles of transmission were preceded by imported cases, predominantly from Italy. We observed an average of 4.8 days (s.d. = 4.0) between the arrival to a Romanian county and COVID-19 confirmation. Furthermore, among the first 147 COVID-19 patients, 88 were imported cases (64 carriers from Italy), 54 were domestic cases, while for five cases the source of infection was unknown. The early human-to-human transmission networks illustrated a limited geographical dispersion, the presence of super-spreaders and the risk of COVID-19 nosocomial infections. COVID-19 occurred in Romania through case importation from Italy. The largest share of the Romanian diaspora is concentrated especially in the northern parts of Italy, heavily affected by COVID-19. Human mobility (including migration) accounts

for the COVID-19 transmission and it should be given consideration while tailoring prevention measures.

## 1. Introduction

The novel coronavirus disease 2019 (COVID-19) pandemic, caused by the Severe Acute Respiratory Syndrome Coronavirus 2 (SARS-CoV-2), originated from Wuhan, China. As of 5 May 2020, COVID-19 has spread to 215 countries, areas and territories, with confirmed cases at 3 517 345 and a reported death toll of 243 401 [1]. Between 26 February and 5 May, Romania has confirmed 13 512 cases and 803 deaths [1]. As a response to the COVID-19 outbreak development, the Romanian authorities have gradually taken several preventive measures [2], such as a 14-day institutionalized quarantine for people travelling from affected regions in Italy (21 February), ban of public gatherings and school closures (8–13 March), 30-day state of emergency (16 March), national lockdown (24 March) and a 30-day state of emergency extension (14 April). In parallel, especially after 15 March 2020, the Romanian authorities have progressively restricted public access to microdata on the novel coronavirus spreading across Romania. In effect, several non-governmental bodies as well as the University of Bucharest, through open letters, protested against this measure. In their protest, it was claimed that, in the early stage of the COVID-19 outbreak in Romania, the availability of public information is critical for the research efforts by independent teams to provide robust evidence to guide effective measures.

In this cross-sectional study, we describe the early-stage dynamics of COVID-19 in Romania (an East European country, with a resident population of 19 414 458 [3] and an administrative organization divided into 41 counties plus Bucharest city (the capital)). Given the lack of studies on this particular country case, our intention is to shed light on how the novel coronavirus entered Romania and on the structural features of the first documented human-to-human transmission networks. Specifically, we profile the first officially confirmed cases (index patients) by administrative counties (also including the capital city, Bucharest), we look at the share of imported cases in the list of the first 147 COVID-19 cases confirmed for Romania and we examine the spread of the SARS-CoV-2 through observed individual-level networks.

Understanding the transmission dynamics in the early stage of COVID-19 outbreaks has been deemed crucial for the assessment of epidemiological situations as well as for setting the effectiveness of outbreak control measures [4]. By reporting the case of Romania and the corresponding datasets, we contribute to the recent efforts of the European Union to develop a joint European Roadmap towards lifting COVID-19 containment measures [5]. Our study is also in line with the rapidly cumulating literature on ascertaining the patterns of the global human mobility COVID-19 transmission [6] and its European as well as global spread features [7–14].

## 2. Material and methods

For this observational, cross-sectional study, epidemiological data were collected from the Romanian Ministry of Health communiqués. Data from official statements were supplemented with information reported by Romanian local media. This strategy was deemed to improve the quality and accuracy of the information communicated by the Romanian officials. The datasets, supporting the analysis and results, are publicly available for consultation [15]. Every reported individual case (microdata) is assigned online public sources which can be subsequently accessed for further details. The level of data granularity prevents any form of disclosing and tracking the infected persons.

We employed the following case selection method: firstly, we started by selecting for each Romanian county, the first patients (index cases). Afterwards, we continued by selecting all publicly available individual cases officially reported on the territory of Romania. When the official Romanian authorities restricted public access to COVID-19 infected patients, we stopped the data collection procedure. Human-to-human transmission networks were built by scanning, in the available official data, for infection chains. The process was driven by the condition that both the source and the target of a chain are officially COVID-19 confirmed cases.

Consequently, the first dataset includes attribute data on 147 cases (i.e. the index patients by each of the Romanian administrative counties and Bucharest city ($n = 47$) as well as the first 100 reported cases), referring to variables, such as sex, age, place of residence (county), probable place and country of infection, the return date to Romania—where appropriate, and the COVID-19 confirmation date. The first confirmed case in the dataset is on 22 February 2020, while the last one is on 2 April 2020). The second dataset refers to human-to-human transmission networks extracted from the official

evidence available since 26 February 2020, as of 20 March 2020. The information refers to variables, such as chains of virus transmission (who to whom transmits SARS-CoV-2), the probable modes of transmitting the virus (i.e. how a patient transmitted SARS-CoV-2 to another patient), location (Romanian county), sex, age, probable place of infection, probable country of infection and the COVID-19 confirmation date.

The data sources for both attribute and network variables were the same as the ones used for COVID-19 case identification (official communiqués supplemented with local media insights). Subsequently, our analysis was performed on a population of officially confirmed cases that corresponds to the early spread of COVID-19 in Romania.

Descriptive statistics on all variables, timeline visual encoding, schematic relations between the return to Romania and COVID-19 confirmation dates, as well as temporal patterns, were performed on the collected epidemiological attribute data. Exploratory network analysis, time-status displays of the novel coronavirus individual level infections and temporal patterns of COVID-19 detection were performed on observed human-to-human transmission networks. We did not apply any missing data imputation technique in either of the two datasets.

## 3. Results

In this section of the paper, we start by reporting the main findings on the first 147 confirmed COVID-19 cases and continue with presenting the results on the profile of the index patients reported in the Romanian counties (including Bucharest city). Unknown information on the cases is reported as missing data. Owing to data access restrictions currently practised by the authorities and to ongoing COVID-19 confinement measures in Romania, it is impossible to perform further research so as to decrease the level of missing data.

Descriptive statistics on the first 147 cases reported in Romania indicate that 60% are imported cases ($n = 88$), from: Italy (64 cases), The United Kingdom (five cases), Israel and Germany (four cases each), Austria, Belgium and France (two cases each), Norway, Poland, Spain, The United Arab Emirates and The United States of America (one case each). For 37%, Romania is the probable infection country ($n = 54$) while for five cases the probable place of infection is unknown. Additionally, 46% are females ($n = 67$) with an average age of 43 years old (s.d. = 14.6, missing = two cases), 53% are males ($n = 78$) with an average age of 41 years old (s.d. = 15.4, missing = four cases). For two cases both sex and age are unknown. The average age in all cases is 41.9 years old (s.d. = 15.0, missing = eight cases).

Using official communiqués and media reports, we document, for Romania, 47 index cases: 18 females—average age: 47.8 (s.d. = 14.8, missing = two cases), 27 males—average age: 47.6 (s.d. = 13.1, missing = four cases) and two of unknown sex. Among the index cases, 44 are imported from: Italy (39 patients: Lombardia—four, Bergamo—three, Trento—three, Milano—two, Ancona/Bologna—one, Cattolica—one, Rimini—one, Roma—one, Udine—one, Venezia—one, and unknown places—21), Great Britain (two patients, London), Germany (one, unknown place) and France (one, unknown place). Considering the distribution of the index cases by Romanian administrative units (including Bucharest city), we observe that 35 counties (83%) have index patients from Italy (three counties have two cases each). Additionally, 39 of the 42 Romanian counties have imported cases and, in only three situations, are there inter-county COVID-19 transmissions (residents of a county contracted SARS-CoV-2 from residents of another county). Figure 1a illustrates, for the 47 index cases, the individual time windows between the arrival to a Romanian county and COVID-19 confirmation, with visual variables indicating nationality, sex and the probable place of infection. On average, there are 4.8 days (s.d. = 4.0, missing = four cases) between the arrival and the COVID-19 confirmation. Figure 1b informs on the detection time dynamics, with a 14-day moving average, showing that Romanian authorities decrease the individual time windows until COVID-19 confirmation especially for index cases who arrive later in Romania. This may reflect how the screening process developed in Romania. The time-lag between arrival (red circles) and COVID-19 confirmation (blue circles) tended to decrease especially after 11 March (figure 1a). Reducing time gaps may show that Romanian authorities improved the COVID-19 case detection with time.

In the human-to-human COVID-19 transmission networks (figure 2a), people are represented as nodes ($n = 159$) while the direction of the novel coronavirus transmission is indicated by arrows ($n = 203$ ties). Visual variables (colours) are marking modes of COVID-19 transmissions: red indicates nosocomial infections of COVID-19 (this is the case for 53 of the individuals represented in the networks), blue—intra-familial transmissions or family clusters (18), magenta—workplace infections (11), green—virus spreading in enclosed public places: high-schools, dance hall, bars and restaurants (7), grey and black—unknown means of transmissions (7). The distribution of the COVID-19 transmission modes is illustrated, using the same colour legend, in figure 2c. In the networks, infection sources or seeds (63

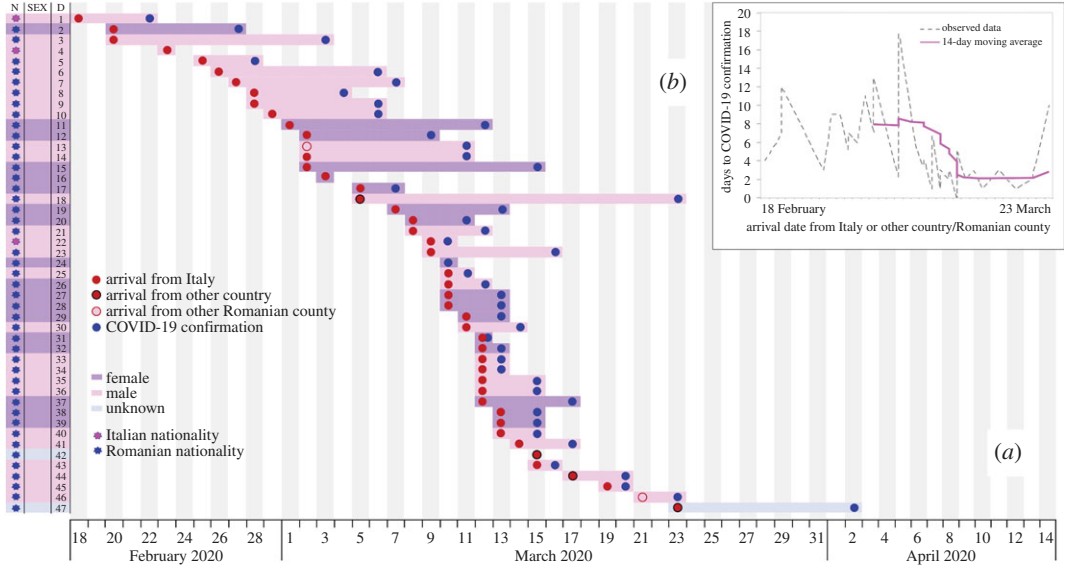

**Figure 1.** Time window between arrival and COVID-19 confirmation for the index cases. (*a*) For index cases, an illustration of individual time windows between the arrival to a Romanian county from Italy (or other country/Romanian county) and the COVID-19 confirmation. (*b*) The time-lag dynamics between arrival and COVID-19 confirmation.

people) are represented as triangles. The longest observed geodesic distance (the shortest distance between any pair of nodes) equals three. It is noteworthy that the networks do not share the same county. The skewed distribution of the out-degree scores (figure 2*b*) indicates that COVID-19 patients transmitted the virus in different degrees or rates. For example, 13 people (8% of all nodes) account for 103 of the total observed transmission arrows (51% of all ties), and two people account for a quarter of all observed ties (super-spreaders)—their corresponding networks are illustrated in figure 2*a*(i) and *a*(ii).

The largest share of the cases reported in the transmission networks (figure 2*a*) is confirmed between 15 and 19 March, i.e. 71 cases, accounting for 45% of all infected people (overall, the first case is confirmed on 22 February, while the last cases on 19 March—not all index cases are included in the observed transmission networks). A time-status display of the COVID-19 infections for the largest two human-to-human transmission networks (exhibited as figure 2*a*(i) and *a*(ii)) is available in figure 2*d*(i) and *d*b. This is indicative for a specific infection pattern (nosocomial). Vertical lines mark the day when a COVID-19 case is confirmed and allow time-stamp comparison against the infection source, i.e. the seed; visual variables in figure 2*d* have the same meaning as in figure 2*a*. The network illustrated as figure 2*d*(i) has 48 nodes: eight females—average age: 40.3 years (s.d. = 13.5), 11 males—average age: 40.1 years (s.d. = 17.8) and 29 people of unknown sex and age. All individuals are residents in the same city. Thirty-five of the individuals contracted the COVID-19 infection in the (same) hospital, seven were of family cluster nature and five workplace cases. The second largest time-displayed network (figure 2*d*(ii)) has 16 nodes: 14 females—average age: 47 years (s.d. = 11.2) and two males of 19 and 4 year olds. All individuals are resident in the same county and contracted the COVID-19 infection in the (same) hospital.

The main results indicate that Romanian travellers, predominantly, from Italy to Romania were the main source of virus spread into the country. Their return gave rise to local COVID-19 human-to-human transmission networks of a limited number of chains. Despite the early stage in their development, these networks embedded super-spreaders (most infections were accounted by a few cases). The most frequent mode of virus transmission in these networks was nosocomial which suggests rather a local and geographically bounded circulation. COVID-19 spread across Romania occurred not by human-to-human inter-city or inter-provinces transmission networks. Independent parallel multiple case imports from abroad in all administrative counties were the main cause.

## 4. Discussion

This is, to our knowledge, the first paper aiming to bring forth how the COVID-19 outbreak occurred in Romania. We showed that, in almost all administrative counties of Romania, index cases were, with few

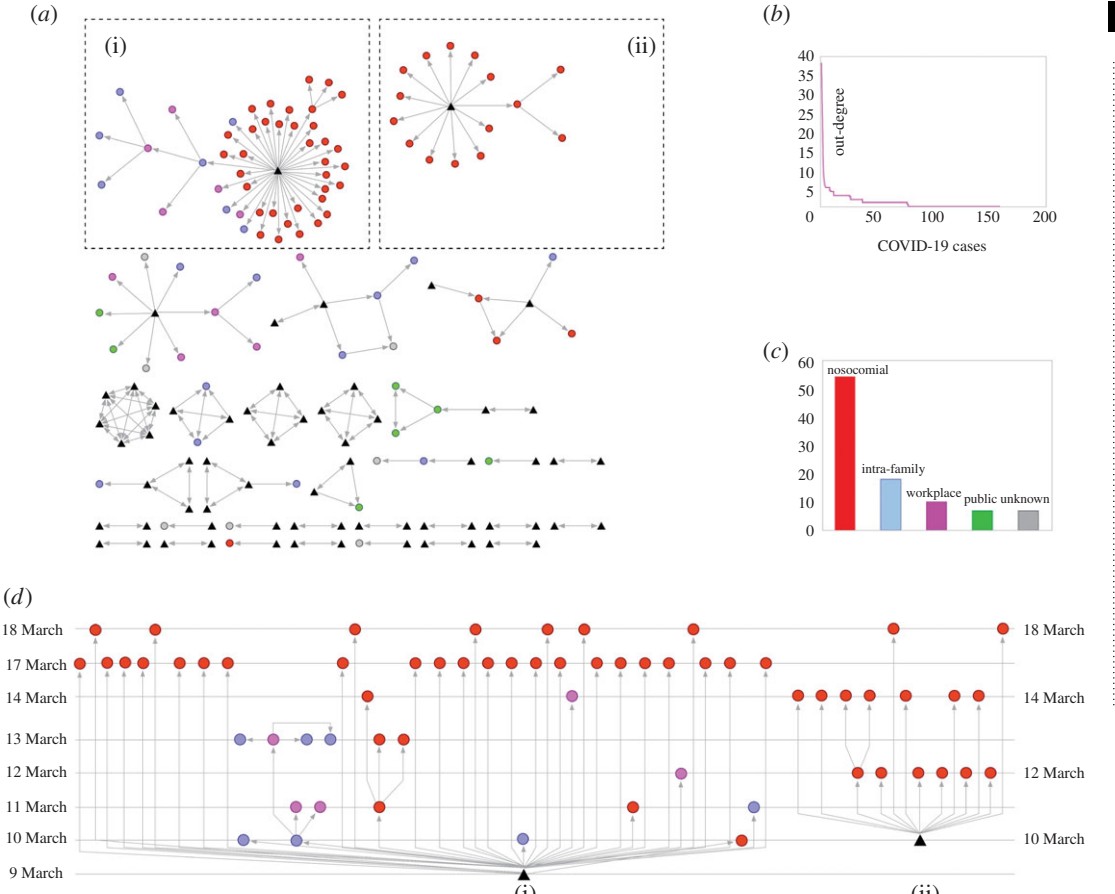

**Figure 2.** Human-to-human COVID-19 transmission networks. (*a*) COVID-19 human-to-human transmission networks: 159 people (nodes) and 203 transmission ties (arrows). Colours indicate how people acquired the COVID-19 infection: nosocomial (red), intra-family (blue), workplace (magenta), green (public places) and grey/black (unknown source). Triangles mark the source of the infection in the networks (the seeds). (*b*) The distribution of the out-degree scores (how many people an individual infected) for all the 159 nodes included in the networks. (*c*) The pattern of COVID-19 transmissions, in absolute scores and by colours corresponding to the networks in (*a*). (*d*) Time-status display of the largest two transmission networks illustrated in (*a*) ((i) and (ii)). Horizontal lines mark temporal order (time-stamps) in days of COVID-19 infection confirmations. Visual variables (colours and shapes) have the same meaning as in (*a*).

exceptions, returning Romanians from Italy. Moreover, of the first 147 reported patients, 60% were imported cases (64 from Italy and 24 worldwide—with three exceptions, all were Romanians). The multiple COVID-19 importations resulted in early local transmission networks with: short paths (maximum three transmission chains), limited geographical dispersion (fragmentation across counties) and high-level of centralization (e.g. two people account for a quarter of all COVID-19 infections, i.e. detection of super-spreaders [16]).

In a nutshell, we suggest that, in Romania, local cycles of transmission were preceded by imported cases, predominantly from Italy. Our research is in line with other previous articles illustrating case imports from Italy, such as Bolivia [17], Brazil [18], Austria, Croatia and Spain [19]. For Romania, this case-importation route is not surprising as Italy has been, for at least the last two decades, the main migration destination for Romanians. Available estimations [20] indicate Romania as the nation with the largest share of migrants in Italy, 23%—1 190 091 people. The majority of Romanian immigrants is located in the north of Italy and, especially, in the COVID-19 heavily affected territories of Lombardia, Emilia-Romagna, Piemonte and Veneto (these four Italian regions account for 47.3% of all Romanians living in Italy). Additionally, all Romanian counties have Italy as a top destination for circular migration, i.e. temporary Romanian workers in a constant back and forth travel movement. The human mobility corridor between Romania and Italy was built during 2000–2010, when it was one of the 10 migration corridors worldwide with the largest average annual increase in international migrants [21]. From this perspective, our study calls for future detailed research on the impact of

human (circular) mobility upon the spread of diseases, in general, and of COVID-19, in particular. This is highlighted by the fact that, unlike other countries [7,9], in Romania, the COVID-19 travellers were returning Romanians and not tourists. Even if more work is needed, the link between COVID-19 transmission and migration [22,23] or human mobility corridors [24] has already been marked.

Our examination of the early local observed transmission networks supports already documented patterns of COVID-19 spread: inside hospitals [25], at workplaces [26], within the families of travelling patients [27] and via carriers (in enclosed public spaces) [28]. The risk of spreading COVID-19 infections in Romanian hospitals illustrated by the early transmission networks is supported by recent official data. For instance, on 19 April, in Romania, the officially reported rate of infection among healthcare workers was of 12.7% (981 healthcare workers to a total of 7707 cases). In Italy (Lombardy region) and Spain, two of the most affected European territories, on 23 April, the rate among healthcare workers was estimated at 20% [29].

We extend the evidence on how bilateral human mobility corridors (e.g. Romania—Italy) account for the COVID-19 transmission. We imply that the resumption of (circular) international migration, as an effect of relaxing travel restrictions, can contribute to a second COVID-19 outbreak. Given this context, we suggest that public health experts and responsible authorities need to prepare plans and strategies to manage the human mobility and especially the circular movement of the temporary workforce between origin and destination places. In addition, we argue that the detection of the super-spreaders (similar to those embedded in the early human-to-human transmission networks) is of vital importance for controlling infection rates.

Our results have implications for understanding the COVID-19 global spread, in the context of human mobility and case importations. This may help inform authorities in their efforts of tailoring mobility control measurements. Further research is needed to increase knowledge on the already documented link between circular migration and COVID-19 transmission. Bi-national human mobility corridors actively connect places of origin and destination, supporting flows of travellers. Insights on the functioning mechanisms of these corridors could support the efforts of public health experts in planning and preparing for different scenarios in later stages of COVID-19 global transmissions.

Currently, contact tracing (human-to-human transmission networks) remains one of the most important tools for managing the COVID-19 pandemic. Any evidence on social contact networks is considered critical for the rapid detection of COVID-19 spread [30]. It has already been proven that the shape and time dynamics of contact networks are essential in predicting disease emergence, spread and persistence [31]. Moreover, it has been shown that observed contact tracing mitigates the virus spread [32] and reduces the size of the simulated outbreaks [33]. In other words, it informs experts in breaking transmission chains and controlling infectious disease outbreaks [30]. Sharing hard to collect raw data on observed human-to-human transmission networks gives insight on high-risk settings (public or shared transportation, workplaces, schools, healthcare settings, etc.). Moreover, it helps scientists to develop mathematical models that are beneficial for testing various transmission scenarios [34]. Our article and its associated raw data are in line with the practice in the field, i.e. it contributes to the modelling of infectious disease parameters [35]. Put differently, it is useful for the mathematical simulations that aim to push the reproductive rate of the virus below specific thresholds. It supports the comparison of simulation results with real data on COVID-19 and contributes to modelling possible counter-measures [36]. Given the dearth of real data on virus transmission, we hope our work will support the current efforts combating against COVID-19.

This study has limitations and the results should be addressed with caution. Firstly, we relied only on the reports of the national authorities and variations exist in the details, the accuracy and the quality of the information communicated to the public. For this reason, in some cases, we had to complement the available official information with local media investigations—especially for the index patients in the Romanian counties. Secondly, the official reports are extremely poor in detail, preventing us from performing any advanced statistical modelling and analysis. Thirdly, given that Romanian authorities ceased reporting individual-level network transmission data, especially after 15 March, we limited the analysis to only early available human-to-human transmission networks. Fourth, the social distancing measures introduced after 19 March, impeded us from any attempt of supplementing data by collecting epidemiological fieldwork information.

Data accessibility. Data and code for the early spread of COVID-19 in Romania can be accessed through the Dryad Digital Repository at: https://doi.org/10.5061/dryad.x69p8czfm [15].

Authors' contributions. All authors contributed equally to the manuscript. M.-G.H, M.P. and J.L. made substantial contributions to the design of the work, to data collection, analysis and interpretation. They also drafted the article and revised it critically. All three authors approved the final version of the manuscript before submission and agreed to be accountable for all aspects of the work in ensuring that questions related to the accuracy or integrity of any part of the research are appropriately investigated and resolved.

Competing interests. We declare we have no competing interests.

Funding. M.-G.H. received funding from The Executive Agency for Higher Education, Research, Development and Innovation Funding (UEFISCDI—Government of Romania) (code: PN-III-P1-1.1-TE-2016-0362). M.P. received funding from The Slovenian Research Agency (grant nos J4-9302, J1-9112 and P1-0403). J.L. received funding from Deutsche Forschungsgemeinschaft (DFG grant no. LE 2237/2-1).

Acknowledgements. For their valuable contribution to this study, we gratefully acknowledge Charles Fletcher (proofreading) and Laura Trandafir (figure formatting preparations).

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
