## [Reviewer comments · Royal Society Open Science]

Review History

RSOS-200780.R0 (Original submission)

Review form: Reviewer 1

Is the manuscript scientifically sound in its present form?

Yes

Are the interpretations and conclusions justified by the results?

Yes

Is the language acceptable?

Yes

Do you have any ethical concerns with this paper?

No

Have you any concerns about statistical analyses in this paper?

No

Recommendation?

Accept as is

Comments to the Author(s)

This paper is an urgent report of COVID-19 in Romania. Although the reported data seems rather raw than intrigued, its importance for helping people seeking modeling should be admired. And the content of data itself is quite valuable.

Amazingly, the authors successfully traced respective personal data of the first 147 cases confirmed COVID-19 in Romania, of which early stage data is visually shown in Fig.1 as the time-window. It obviously proved that most of the cases can be said as 'imported' from Italy. Another important point can be drawn from Fig. 1, although the authors did not address so much, is that for the very early period, relatively long incubation period observed in Romania, denoted by the gap of red and blue circles, was observed. Yet, After 11th March, such gap became small.

Although this gap does not directly mean what-is-called 'incubation period' from epidemiological standpoint, it may mean a reflection of the screening process in Romania being well prepared with time preceding. Yet, it seems quite interesting thing. Moreover, I can admire the authors' great effort to correct personal but important data.

Fig. 2 shows human-to-human transmission network in Romania, which proves that COVID-19 quickly spread in the country though domestic transmission network after the initial period of importation predominantly from Italy.

Referring to the general statement that we, as scientists, should correct and share any information concerning COVID-19 and SARS-CoV-2, especially first urgent data in respective countries, I strongly suggest this MS should be welcomed to the journal.

One suggestion so as to improve this MS to be more impressive to wider readership is that the authors should add a certain discussion of how their first report can contribute and trigger following studies, especially modelling effort to understand and combat with COVID-19.

Review form: Reviewer 2 (Jan Korbel)

Is the manuscript scientifically sound in its present form?

Yes

Are the interpretations and conclusions justified by the results?

Yes

Is the language acceptable?

Yes

Do you have any ethical concerns with this paper?

No

Have you any concerns about statistical analyses in this paper?

No

Recommendation?

Accept as is

Comments to the Author(s)

The study is a timely contribution to the important topic of the early spreading of COVID and how to prevent it. The authors analyzed the first cases of coronavirus spread in Romania and analyzed the transmission networks. The dataset attached to the paper can be used for comparison effectivity of government measures in different countries. I suggest accepting the paper in the current form.

Review form: Reviewer 3

Is the manuscript scientifically sound in its present form?

No

Are the interpretations and conclusions justified by the results?

No

Is the language acceptable?

Yes

Do you have any ethical concerns with this paper?

No

Have you any concerns about statistical analyses in this paper?

No

Recommendation?

Major revision is needed (please make suggestions in comments)

Comments to the Author(s)

1. The figures on last two pages are not valid, please correct them for a better comprehension of this paper.
2. The methods used in the paper is too simple, I could not see any novelty from the methodology part. In other words, I could not see value other than the statistical results obtained from the data.
3. The results and findings are not very insightful. More discussion is expected.

Please re-prepare the draft and address the above concerns.

Decision letter (RSOS-200780.R0)

Dear Dr Hancean,

The editors assigned to your paper ("Early spread of COVID-19 in Romania: imported cases from Italy and human-to-human transmission networks") have now received comments from reviewers. We would like you to revise your paper in accordance with the referee and Associate Editor suggestions which can be found below (not including confidential reports to the Editor). Please note this decision does not guarantee eventual acceptance.

Please submit a copy of your revised paper before 17-Jul-2020. Please note that the revision deadline will expire at 00.00am on this date. If we do not hear from you within this time then it will be assumed that the paper has been withdrawn. In exceptional circumstances, extensions may be possible if agreed with the Editorial Office in advance. We do not allow multiple rounds of revision so we urge you to make every effort to fully address all of the comments at this stage. If deemed necessary by the Editors, your manuscript will be sent back to one or more of the original reviewers for assessment. If the original reviewers are not available, we may invite new reviewers.

- Data accessibility

If you wish to submit your supporting data or code to Dryad (<http://datadryad.org/>), or modify your current submission to dryad, please use the following link:
<http://datadryad.org/submit?journalID=RSOS&manu=RSOS-200780>

- Competing interests

- Authors' contributions

- Acknowledgements

- Funding statement

Kind regards,

Andrew Dunn

on behalf of Dr Derek Abbott (Associate Editor)

Comments to Author:

Reviewers' Comments to Author:

Reviewer: 1

Comments to the Author(s)

This paper is an urgent report of COVID-19 in Romania. Although the reported data seems rather raw than intrigued, its importance for helping people seeking modeling should be admired. And the content of data itself is quite valuable.

Amazingly, the authors successfully traced respective personal data of the first 147 cases confirmed COVID-19 in Romania, of which early stage data is visually shown in Fig.1 as the time-window. It obviously proved that most of the cases can be said as 'imported' from Italy. Another important point can be drawn from Fig. 1, although the authors did not address so much, is that for the very early period, relatively long incubation period observed in Romania, denoted by the gap of red and blue circles, was observed. Yet, After 11th March, such gap became small.

Although this gap does not directly mean what-is-called 'incubation period' from epidemiological standpoint, it may mean a reflection of the screening process in Romania being well prepared with time preceding. Yet, it seems quite interesting thing. Moreover, I can admire the authors' great effort to correct personal but important data.

Fig. 2 shows human-to-human transmission network in Romania, which proves that COVID-19 quickly spread in the country through domestic transmission network after the initial period of importation predominantly from Italy.

Referring to the general statement that we, as scientists, should correct and share any information concerning COVID-19 and SARS-CoV-2, especially first urgent data in respective countries, I strongly suggest this MS should be welcomed to the journal.

One suggestion so as to improve this MS to be more impressive to wider readership is that the authors should add a certain discussion of how their first report can contribute and trigger following studies, especially modelling effort to understand and combat with COVID-19.

Reviewer: 2

Comments to the Author(s)

The study is a timely contribution to the important topic of the early spreading of COVID and how to prevent it. The authors analyzed the first cases of coronavirus spread in Romania and

analyzed the transmission networks. The dataset attached to the paper can be used for comparison effectivity of government measures in different countries. I suggest accepting the paper in the current form.

Reviewer: 3

Comments to the Author(s)

1. The figures on last two pages are not valid, please correct them for a better comprehension of this paper.
2. The methods used in the paper is too simple, I could not see any novelty from the methodology part. In other words, I could not see value other than the statistical results obtained from the data.
3. The results and findings are not very insightful. More discussion is expected.

Please re-prepare the draft and address the above concerns.

Author's Response to Decision Letter for (RSOS-200780.R0)

See Appendix A.

Decision letter (RSOS-200780.R1)

Dear Dr Hancean,

It is a pleasure to accept your manuscript entitled "Early spread of COVID-19 in Romania: imported cases from Italy and human-to-human transmission networks" in its current form for publication in Royal Society Open Science. The comments of the reviewer(s) who reviewed your manuscript are included at the foot of this letter.

on behalf of Dr Derek Abbott (Associate Editor)
openscience@royalsociety.org

Follow Royal Society Publishing on Twitter: [@RSocPublishing](https://twitter.com/RSocPublishing)
Follow Royal Society Publishing on Facebook:
<https://www.facebook.com/RoyalSocietyPublishing.FanPage/>
Read Royal Society Publishing's blog: <https://blogs.royalsociety.org/publishing/>

Appendix A

Manuscript ID: RSOS-200780

Manuscript title: "Early spread of COVID-19 in Romania: imported cases from Italy and human-to-human transmission networks" (authors: Marian-Gabriel Hâncean, Matjaž Perc, Jürgen Lerner)

The Royal Society Open Science journal

June, 28, 2020

Dear referees,

We are grateful for the time and effort you devoted to reading and assessing our manuscript. Your comments and suggestions helped us to prepare an improved version of the manuscript. We have done our best to fully address all the comments, comprehensively and with love to detail. As an overview of the changes reflecting referees' feedback, we mark that we added two new paragraphs, *i.e.*, one new paragraph in the *Results* section and one new paragraph in the *Discussion* section.

In what follows, we respond to every point raised by each of the three referees. Our response letter contains the complete set of reviews (written in blue) with our responses interleaved (written in black). We strived to be clear about what we changed relative to the previous version. We hope that our revised manuscript will meet your expectation and be recommended for publication in The Royal Society Open Science journal.

Referee #1

Comment: This paper is an urgent report of COVID-19 in Romania. Although the reported data seems rather raw than intriguing, its importance for helping people seeking modeling should be admired. And the content of data itself is quite valuable.

Our response: We are thankful to the Referee for considering the data in our manuscript as valuable and for stressing its importance in the context of COVID-19 in Romania.

Comment: Amazingly, the authors successfully traced respective personal data of the first 147 cases confirmed COVID-19 in Romania, of which early stage data is visually shown in Fig.1 as the time-window. It obviously proved that most of the cases can be said as 'imported' from Italy. Another important point can be drawn from Fig. 1, although the authors did not address so much, is that for the very early period, relatively long incubation period observed in Romania, denoted by the gap of red and blue circles, was observed. Yet, after 11th March, such gap became small. Although this gap does not directly mean what-is-called 'incubation period' from epidemiological standpoint, it may mean a reflection of the screening process in Romania being well prepared with time preceding. Yet, it seems quite interesting thing. Moreover, I can admire the authors' great effort to correct personal but important data.

Fig. 2 shows human-to-human transmission network in Romania, which proves that COVID-19 quickly spread in the country through domestic transmission network after the initial period of importation predominantly from Italy.

Our response: We thank the Referee for the interpretation as well as for the implicit suggestion. We believe the suggestion is a valuable one. In effect, we brought into the light the observation referring to the gaps between red and blue circles (illustrated in Fig. 1). Staying in line with the suggestion made by the Referee, we added a new paragraph in the *Results* section. This new paragraph is copied verbatim below:

"This may reflect how the screening process developed in Romania. The time-lag between arrival (red circles) and COVID-19 confirmation (blue circles) tended to decrease especially after March, 11 (Figure 1A). Reducing time gaps may show that Romanian authorities improved the COVID-19 case detection with time preceding." (page 4 in the revised manuscript)

Comment: Referring to the general statement that we, as scientists, should correct and share any information concerning COVID-19 and SARS-CoV-2, especially first urgent data in respective countries, I strongly suggest this MS should be welcomed to the journal.

Our response: We thank the Referee for the careful reading of our manuscript as well as for the encouraging and constructive feedback. Moreover, we are glad that the Referee strongly suggests our manuscript should be welcomed to the journal.

Comment: One suggestion so as to improve this MS to be more impressive to wider readership is that the authors should add a certain discussion of how their first report can contribute and trigger following studies, especially modelling effort to understand and combat with COVID-19.

Our response: We strongly agree with this suggestion. In effect, we introduced a new paragraph to better reflect how the manuscript's content and its associated raw data may prove helpful for scientists in their efforts of developing tools and studies to understand and combat COVID-19. Consequently, in *the Discussion section* of the manuscript we placed the following new text:

“Currently, contact tracing (human-to-human transmission networks) remains one of the most important tools for managing COVID-19 pandemic. Any evidence on social contact networks is considered critical for the rapid detection of COVID-19 spread (1). It has already been proven that the shape and time dynamics of contact networks are essential in predicting disease emergence, spread and persistence (2). Moreover, it has been shown that observed contact tracing mitigates the virus spread (3) and reduces the size of the simulated outbreaks (4) In other words, it informs experts in breaking transmission chains and controlling infectious disease outbreaks (1). Sharing hard to collect raw data on observed human-to-human transmission networks gives notice on high risk settings (public or shared transportation, workplaces, schools, healthcare settings etc.). Moreover, it helps scientists to develop mathematical models that are beneficial for testing various transmission scenarios (5). Our article and its associated raw data are in line with the practice in the field, i.e., it contributes to the modelling of infectious disease parameters (6) Put it differently, it is useful for the mathematical simulations that aim to push the reproductive rate of the virus below specific thresholds. It supports comparison of simulation results with real data on COVID-19 and contributes to modelling possible counter-measures (7). Given the dearth of real data on virus transmission, we hope our work will support the current efforts combating against COVID-19.” (page 6 in the revised manuscript)

The references associated to the new paragraph are as follows:

1. World Health Organization. Contact tracing in the context of COVID-19 [Internet]. World Health Organization; 2020 [cited 2020 Jun 28]. Available from: <https://www.who.int/publications/i/item/contact-tracing-in-the-context-of-covid-19>
2. Moreno Y, Pastor-Satorras R, Vespignani A. Epidemic outbreaks in complex heterogeneous networks. *Eur Phys J B*. 2002 Apr;26(4):521–9.
3. Ish P, Agrawal S, Goel AD, Gupta N. Contact tracing: Unearthing key epidemiological features of COVID-19. *SAGE Open Med Case Rep*. 2020 Jan;8:2050313X2093348.
4. Kucharski AJ, Klepac P, Conlan AJK, Kissler SM, Tang ML, Fry H, et al. Effectiveness of isolation, testing, contact tracing, and physical distancing on reducing transmission of SARS-CoV-2 in different settings: a mathematical modelling study. *Lancet Infect Dis*. 2020 Jun;S1473309920304576.
5. Hellewell J, Abbott S, Gimma A, Bosse NI, Jarvis CI, Russell TW, et al. Feasibility of controlling COVID-19 outbreaks by isolation of cases and contacts. *Lancet Glob Health*. 2020 Apr;8(4):e488–96.
6. Hens N, Shkedy Z, Aerts M, Faes C, Van Damme P, Beutels P. Modeling Infectious Disease Parameters Based on Serological and Social Contact Data: A Modern Statistical Perspective [Internet]. New York, NY: Springer New York; 2012 [cited 2020 Jun 28]. (Statistics for Biology and Health; vol. 63). Available from: <http://link.springer.com/10.1007/978-1-4614-4072-7>
7. Giordano G, Blanchini F, Bruno R, Colaneri P, Di Filippo A, Di Matteo A, et al. Modelling the COVID-19 epidemic and implementation of population-wide interventions in Italy. *Nat Med*. 2020 Jun;26(6):855–60.

Referee #2

Comment: *The study is a timely contribution to the important topic of the early spreading of COVID and how to prevent it. The authors analyzed the first cases of coronavirus spread in Romania and analyzed the transmission networks. The dataset attached to the paper can be used for comparison of the effectiveness of government measures in different countries. I suggest accepting the paper in the current form.*

Our response: We deeply appreciate the comprehensive and positive assessment of our Referee. We are delighted our manuscript has been recommended for publication in present form.

Referee #3

Comment: *1. The figures on last two pages are not valid, please correct them for a better comprehension of this paper.*

Our response: We have carefully checked both figures on the last two pages, and we have found nothing that would make them "not valid".

Comment: *2. The methods used in the paper is too simple, I could not see any novelty from the methodology part. In other words, I could not see value other than the statistical results obtained from the data.*

Our response: Papers in The Royal Society Open Science journal do not require methodological advances for publication. We also disagree that our methods are "too simple". Moreover, the Royal Society Open Science journal has committed to rapidly assessing and making available research as well as raw data on COVID-19 pandemic worldwide. This initiative supports the efforts of international academic community in collecting data from all countries affected by COVID-19 and openly share them for increasing the pace in fighting against virus spread. On top of that, it was explicitly stated in the manuscript (pages 1-2, 7), that the availability of raw data was heavily limited. Our methodology had been devised in accordance with the manuscript's objectives and available information. We strongly believe that we are in line with the World Health Organization as well as with the observations made by Referees 1 and 2 when we emphasize that any piece of evidence and research concerning COVID-19 spread, irrespective the country, should be timely made available to the general audience and to the international research community.

Comment: *3. The results and findings are not very insightful. More discussion is expected.*

Our response: We disagree. Our findings are as follows:

- Romanians travellers, predominantly, from Italy to Romania were the main source of virus spread into the country
- The return of Romanian travellers gave rise to local COVID-19 human-to-human transmission networks of limited number of chains
- Despite the early stage in their development, transmission networks embedded super-spreaders (most infections were accounted by a few cases).
- The most frequent mode of virus transmission in these networks was nosocomial which suggests rather a local and geographically bounded circulation.
- COVID-19 spread across Romania occurred not by human-to-human inter-city or inter-provinces transmission networks
- Independent parallel multiple case imports from abroad in all administrative counties were the main cause of the COVID-19 outbreak in Romania.

Our conclusion was that, in almost all administrative counties of Romania, index cases were, with few exceptions, returning Romanians from Italy. Moreover, of the first 147 reported patients, more than two thirds were imported cases (64 from Italy and 24 worldwide - with three exceptions, all were Romanians). The multiple COVID-19 importations resulted in early local transmission networks with: short paths (maximum three transmission chains), limited geographical dispersion (fragmentation across counties), and high-level of centralization (e.g., two people account for a quarter of all COVID-19 infections, i.e., detection of super-spreaders). In a nutshell, we suggested that, in Romania, local cycles of transmission were preceded by imported cases, predominantly from Italy.